# COVID-19 Pandemic and Helsinki University Hospital Personnel Psychological Well-Being: Six-Month Follow-Up Results

**DOI:** 10.3390/ijerph18052524

**Published:** 2021-03-04

**Authors:** Tanja Laukkala, Jaana Suvisaari, Tom Rosenström, Eero Pukkala, Kristiina Junttila, Henna Haravuori, Katinka Tuisku, Toni Haapa, Pekka Jylhä

**Affiliations:** 1Department of Psychiatry, University of Helsinki and Acute Psychiatry and Consultations, HUS Helsinki University Hospital, 00029 Helsinki, Finland; katinka.tuisku@hus.fi (K.T.); pekka.jylha@hus.fi (P.J.); 2Finnish Institute for Health and Welfare, Mental Health Team, 00271 Helsinki, Finland; jaana.suvisaari@thl.fi (J.S.); henna.haravuori@thl.fi (H.H.); 3Department of Psychology and Logopedics, Faculty of Medicine, University of Helsinki, 00014 Helsinki, Finland; tom.rosenstrom@helsinki.fi; 4Faculty of Social Sciences, Tampere University, 33100 Tampere, Finland; eero.pukkala@cancer.fi; 5HUS Helsinki University Hospital, Nursing Research Center, University of Helsinki, 00029 Helsinki, Finland; kristiina.junttila@hus.fi (K.J.); toni.haapa@hus.fi (T.H.)

**Keywords:** COVID-19 pandemic, hospital personnel, potentially traumatic event, psychological distress

## Abstract

The COVID-19 pandemic has caused an unequally distributed extra workload to hospital personnel and first reports have indicated that especially front-line health care personnel are psychologically challenged. A majority of the Finnish COVID-19 patients are cared for in the Helsinki University Hospital district. The psychological distress of the Helsinki University Hospital personnel has been followed via an electronic survey monthly since June 2020. We report six-month follow-up results of a prospective 18-month cohort study. Individual variation explained much more of the total variance in psychological distress (68.5%, 95% CI 65.2–71.9%) and negative changes in sleep (75.6%, 95% CI 72.2–79.2%) than the study survey wave (1.6%, CI 0.5–5.5%; and 0.3%, CI 0.1–1.2%). Regional COVID-19 incidence rates correlated with the personnel’s psychological distress. In adjusted multilevel generalized linear multiple regression models, potentially traumatic COVID-19 pandemic-related events (OR 6.54, 95% CI 5.00–8.56) and front-line COVID-19 work (OR 1.81, 95% CI 1.37–2.39) was associated with personnel psychological distress but age and gender was not. While vaccinations have been initiated, creating hope, continuous follow-up and psychosocial support is still needed for all hospital personnel.

## 1. Introduction

In Europe, the second wave of the COVID-19 pandemic was rising towards the end of the year 2020. Globally, COVID-19 caused 2,421,075 deaths by 17 February 2021 [1]. By February 2021, Finland (population of 5.5 million) has 49,165 confirmed COVID-19 infections and 708 COVID-19-related deaths (median age 84 years) [2] and a majority of the Finnish COVID-19 pandemic patients are cared for in the Helsinki University Hospital (HUS), serving the Uusimaa region in southern Finland. COVID-19 vaccinations began 27 December 2020 in Finland and the aim is to vaccinate the whole population (those without medical contraindications and willing to take the vaccination).

During the fall and winter 2020 the Finnish population were encouraged to participate in COVID-19 testing if even a single mild flu-like symptom was present and some asymptomatic at-risk groups were tested, such as close-ones of COVID-19 patients, those exposed and travelers crossing the borders. All hospital personnel have easy access to testing. The number of COVID-19 tested still varies, which affects incidence rates. The Uusimaa region COVID-19 testing capacity has been adequate through fall and winter 2020.

During the second wave of the COVID-19 pandemic, a part of the health care and other hospital personnel is again severely challenged, while the pandemic-related workload is unevenly distributed. A prospective cohort study with over 4800 responders (19% of the HUS personnel) began in June 2020 and assessed HUS personnel psychological symptoms and well-being monthly via an electronic survey, which also provides information on psychosocial support services [3].

Front-line, but also other HUS personnel, were psychologically distressed as compared to the general Finnish population in June 2020 [3]. These effects appeared to be mediated via potentially traumatic COVID-19-related events (PTEs), as especially the front-line nursing staff was under pressure and psychologically distressed. At the same time, tailored psychosocial support services have been actively offered to HUS personnel since June 2020 in addition to regular occupational health care services, to support the ability to work and function, which was also followed at a symptom level via this cohort study during the COVID-19 pandemic. Among the hospital personnel, risk of exposure is an issue [3,4] and also a potential source of worry. 

An Israelian cross-sectional study on hospital worker (N = 1570) anxiety during the COVID-19 pandemic revealed that pandemic work in itself (working in a hospital treating COVID-19 patients or in a hospital without COVID-19 patients) was not necessarily associated with anxiety [5]. While some individuals are more prone to anxiety symptoms, individuals prone to COVID-19-related fear differed from anxiety-prone individuals in the German general population [6]. Risk perception of COVID-19 infection—risk of contracting, fear, and perceived threat of COVID-19—negatively correlated with psychological well-being in a Polish healthcare personnel study and coping strategies mediated the relationship between risk perception and psychological well-being [7].

As the second wave of the COVID-19 pandemic challenges hospital personnel again, a recent Cochrane review concludes that there is a lack of evidence on how to support front-line personnel during and after the pandemic and there is a lack of follow-up studies [8,9]. The aim of the present study is to follow the impact of the temporally changing COVID-19 pandemic situation on hospital personnel’s psychological well-being by assessing psychological distress, sleep, PTEs and posttraumatic symptoms [3].

## 2. Materials and Methods

### 2.1. Study Data

This study was approved by the HUS Ethical Committee (6.5.2020, HUS/1488/2020) and permission to conduct the study was obtained from the Joint Authority of the Helsinki and Uusimaa Hospital District (1.6.2020 HUS/157/2020). Since our earlier report [3], new responders (N = 61) have joined the study. This follow-up study is based on the original responders. The available answers span the period from 4 June 2020 to 27 November 2020, totaling 176 days. The baseline survey (wave 0) was open from 4 June to 26 June 2020, follow-up survey wave 1 on 3–31 July, wave 2 on 7–28 August, wave 3 on 4–25 September, wave 4 on 2–23 October, and wave 5 on 6–27 November. Although only 46.8% of the subjects participated in the first follow-up survey wave, 75.7% participated in at least one of the five follow up waves (Table 1). A complete breakdown of missing data patterns is available as an online supplement (Appendix A).

The electronic survey consists of a few open questions about work and possible changes due to the pandemic. There were also four separate questions on COVID-19 pandemic-related potentially traumatic events (PTEs during the COVID-19 pandemic) that were coded yes/no.
(1)Has your work with suspected or confirmed COVID-19 patients included exceptionally disturbing or distressing assignments?(2)Have you had strong anxiety due to your own or a close one’s risk of contracting serious illness for your work with suspected or confirmed COVID-19 patients?(3)Have you or your close one contracted a hospital care requiring severe COVID-19?(4)Has a close one to you died of COVID-19?

Symptom self-rating scales included Primary Care Post-Traumatic Stress Disorder Scale (PC-PTSD-5), Mental Health Index (MHI-5) and Insomnia Severity Index (ISI). MHI-5 asks the following five questions: how much of the time in the last month the responder had considered himself to be a very nervous person, had felt downhearted, had felt calm and peaceful, had felt so down in the dumps that nothing could cheer him up, or considered himself to be a happy person. Raw scores were transformed to a rating between 0 and 100 points. MHI-5 rates under 53 points refer to clinically significant psychological distress [10,11]. In a Finnish general population study, MHI-5 showed good internal consistency and independent factor structure and item characteristics [12]. ISI assesses sleeping problems in a scale from 0 to 28 points; 15 points or over refer to moderate or severe insomnia symptoms [13]. ISI has been sensitive to changes in clinical insomnia studies [13,14,15]. PC-PTSD-5 assesses flashbacks, avoidance, hyperarousal, numbness, and negative cognitive emotions. Three or more “yes” answers to five symptom questions after a potentially traumatic event refer to an elevated risk of post-traumatic stress disorder (PTSD) [16,17]. Cronbach’s alphas were (in the T0 sample) for the different scales: MHI-5 alpha = 0.895, ISI alpha = 0.895, and PC-PTSD-5 alpha = 0.647. The scales correlate significantly. Pearson’s correlation co-efficient varies from 0.449 to 0.581, suggesting that all the scales measure some aspect of psychological distress, i.e., suggesting concurrent validity.

### 2.2. Other Data

Besides the HUS employees, we draw from an open data repository the weekly COVID-19 incidence rates in the Uusimaa region that the HUS hospital district serves. These official numbers by Finnish Institute of Health and Welfare are available from https://sampo.thl.fi/pivot/prod/en/epirapo/covid19case/ (accessed on 4 January 2021).

### 2.3. Statistics

Boxplots and local regression lines were drawn with R version 4.0.2 (22 June 2020) to illustrate the data. Multilevel (mixed-effects) ordinary (Gaussian) and logistic (Binomial) regression models were fitted with lme4 R package, version 1.1-23. A random intercept was used to model employee specific risks that stay constant over repeated measurements. Effects of time were modeled with polynomials of standardized time (calendar dates), and in a sensitivity analysis, with time-specific local COVID-19 log-incidence per week. More specifically, incidence rate *r* was transformed to log(*r* + 1) for regression modeling to linearize the generally exponential growth rate of infection transmission. Our modeling strategy used all available data waves per employees despite him/her missing some of the waves [18,19]. Each respondent contributed to our analysis in proportion to the data they provided and none who had provided some data were excluded from analysis (as some “complete-case analyses” might do). However, we did not try to model the counterfactual situation of how the results would have looked had we had all the data also from employees that did not answer every time but did some of the time.

## 3. Results

### 3.1. Sample Characteristics

Table 1 characterizes the available sample by the survey wave. The waves approximately reflect the time the participants answered the online questionnaire, with some variation in response dates within the waves (Figure 1a). Besides large variations in mental health between individuals (Figure 1b), a mean-level pattern across the waves was evident (Figure 1c) that closely inverse-tracked the local progression of COVID-19 incidence (Figure 1d).

### 3.2. Multilevel Models

Between-individual variation explained much more of the total variance in MHI-5 score (68.5%, 95% CI 65.2–71.9%) and in ISI score (75.6%, 95% CI 72.2–79.2%) than the study wave explained (1.6%, 95% CI 0.5–5.5%; and 0.3%, 95% CI 0.1–1.2%, respectively) in a simple linear multilevel model. This quantifies what we observed, e.g., in Figure 1b: while waves are associated with MHI-5 scores, between-employee variability within waves is much greater in magnitude.

Although nursing staff membership predicted MHI-5 screen for higher distress in the survey wave 0 data [3], this association did not remain significant over the entire follow-up (OR 1.44, 95% CI 0.97–2.14, *p* = 0.073).

We inquired Potentially Traumatic Events (PTEs) and direct COVID-19 patient care at each survey wave, and they predicted the screen for low mental health over the follow up (Table 2), with a magnitude comparable to previous survey wave 0 findings [3]. Three polynomial terms for the response date were significant predictors of low mental health too, likely reflecting the local temporal progression of the COVID-19 pandemic in Finland. Figure 2 illustrates the effects of time and PTE covariates on the probability of low mental health for an employee without any employee-specific vulnerability or resilience (i.e., with zero random effect; note that the employee-specific effects represented majority of variance in MHI-5). This variation in psychological distress according to the epidemic situation has also been seen in the Citizen’s Pulse survey, a monthly general population survey of the Finnish general population which measures people’s attitudes and worries related to the COVID-19 pandemic and measures taken to mitigate its effects (https://www.stat.fi/tup/htpalvelut/tutkimukset/kansalaispulssi_en.html (accessed on 4 January 2021); please see also Appendix A).

We further verified that the local COVID-19 log-incidence predicted low mental health even when adjusting for directly treating COVID-19 patients and experiencing a traumatic event (Model 3 in Table 2), and that our time-covariates captured the effects of log-incidence (Model 4 in Table 2). In addition to the above discussed models, we noted that changes in work due to COVID-19 were associated with low mental health slightly more strongly than COVID-19 patient contact but less strongly than a traumatic event (i.e., with OR 2.97, 95% CI 2.33–3.79).

## 4. Discussion

This study adds information on psychological effects of the temporally changing COVID-19 pandemic situation on hospital personnel. While initial reports from Europe and worldwide have emphasized front-line personnel distress, assessed with cross-sectional surveys on psychological distress, sleep, depression, PTSD symptoms and anxiety, follow-up information is scarce [8,9,20,21]. To our knowledge there are no other prospective cohort studies on hospital personnel psychological well-being during the COVID-19 pandemic in Finland. A cross-sectional survey on anxiety in an area of low COVID-19 incidence revealed no increase in general anxiety assessed by GAD-7 [21] while a Norwegian cross-sectional study from the spring 2020 is in line with our baseline results [3,22].

In our cohort, individual variation explained much of the psychological distress. Reported increase in COVID-19 incidence rates in the Uusimaa district in Finland is associated with psychological distress and negative changes in sleep. COVID-19 second wave here reflects the reported Uusimaa region COVID-19 incidence. COVID-19 infections are reported to the National Infectious Diseases Register in Finland, and on-line reports are compiled by the National Institute of Health and Welfare (see https://experience.arcgis.com/experience/92e9bb33fac744c9a084381fc35aa3c7 (accessed on 4 January year)). On an individual level, PTEs, other life events, resilience and coping strategies may mediate these changes [3,7].

Increase in Uusimaa area COVID-19 incidences appeared at the same time with lower self-reported psychological well-being and increase in sleeping problems among the cohort of our study (please see Figure 1), and a U-curve similar to our data was present in the self-reported stress symptoms of the Finnish general population (Appendix A). In the U.S., initial psychosocial emotional responses assessed via social media were followed by a plateau between March and May 2020, compared to spring 2019 [23], but the Finnish data demonstrate that a second Covid-19 wave can imply a second mental health hit to the population (i.e., habituation has limits and the timely situation matters a lot).

Caring for COVID-19 patients at each survey wave was associated with psychological distress over a six-month time. Although belonging to nursing staff at survey wave 0 was associated with an MHI-5 screen for low mental health in June 2020, this association did not remain significant over the entire six-month follow-up. PTEs were associated with the screen for low mental health over the entire follow-up.

Recognition of distress-prone individuals might help to organize psychosocial support services more effectively during the COVID-19 pandemic. In our cohort, age and gender did not explain psychological distress. While the systematic review of Carnassi et al. (2020) noticed that younger age might predict posttraumatic symptoms among the health care personnel during the COVID-19 pandemic, the effects of gender vary between studies [3,5,20,24].

Temporary insomnia symptoms are common, and often non-pathological responses to environmental, circadian, physiological, and emotional challenges. Instead, chronic insomnia can have adverse effects on health, mental wellbeing and coping with stress. Long-term cardiovascular, neurological, and psychological effects of disrupted sleep are mediated by several mechanisms, for example by low grade inflammation, autonomous hyperarousal, endocrine dysregulation, deficient brain drainage, poor restoration of mitochondrial energy resources, deteriorated memory consolidation and emotion regulation, daytime tiredness, lowered mood, anxiety, and maladaptive behavioral responses [25,26]. Vulnerability to stress-induced sleep disruption is individually varied and related to epigenetic dysregulation of hypothalamic-pituitary-adrenal axis. The associations between insomnia and emotional stress reactivity are bidirectional; unmanageable stress leading to insomnia via hyperarousal, and fragmented sleep exacerbating daytime distress via emotional dysregulation [27,28,29].

Limitations of the study include the fact that the initial responder group represents 19% of Helsinki University Hospital personnel and some of the initial responders (please see the Appendix A), did not continue to fill in the follow-up surveys. Surveys in general have limitations compared to register-based data, and a large number of employees cannot be reached without an electronic survey design in the middle of the pandemic work. Increase in the national COVID-incidence increased return to the survey follow-ups. Liability to psychiatric symptoms strongly reflects individual differences in latent risk factors, including genes and general behavioral traits, e.g., [30,31], and it is not surprising that the same held in our employee data. A large consortium currently seeks to understand the reasons and structures behind individual differences in psychiatric symptoms [32]. Future research on mental-health in the context of Covid-19 might benefit from a more accurate modeling of psychiatric history and comorbidity. Personality might have a role in the vulnerability, resilience, and expressions associated with traumatic events [33], but we were not able to further study this aspect as no personality questionnaire was used.

If there is a selection in our study respondents, we expect that those who work in the health care fields affected by the COVID-19 pandemic choose to continue in this survey. We consider social desirability bias unlikely since participation is voluntary. In line with international position papers on the COVID-19 pandemic, mental health research is needed [4,34,35], as we continue the prospective hospital employee cohort follow-up. Mapping both individual and systemic vulnerability and resiliency factors in this prolonged situation for health care workers should be addressed.

## 5. Conclusions

Recognition of potentially traumatic events during and after the COVID-19 pandemic might help to direct psychosocial support services effectively to psychologically distressed hospital personnel. All such effects of the COVID-19 pandemic cannot be mitigated by employers and psychological distress varied between individuals and also varied in line with the reported COVID-19 incidence, which affects workloads in healthcare settings.

## Figures and Tables

**Figure 1 ijerph-18-02524-f001:**
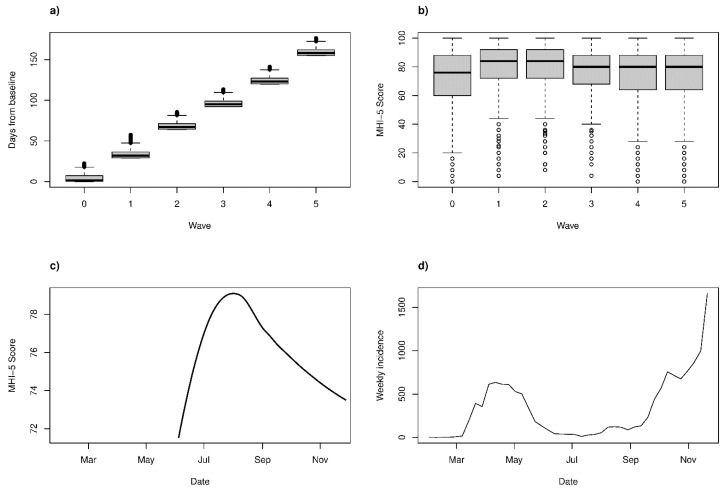
(**a**) Boxplot of days since baseline (first possible day to respond was 4 June 2020) by study wave. (**b**) A 5-item Mental Health Index (MHI-5) by study wave. (**c**) Local regression of MHI-5 score on the exact response date. (**d**) Weekly incidence of COVID-19 locally (in the Uusimaa region, total population ~1.7 million). The incidence data is from an official open-access record by National Institute of Health and Welfare in Finland.

**Figure 2 ijerph-18-02524-f002:**
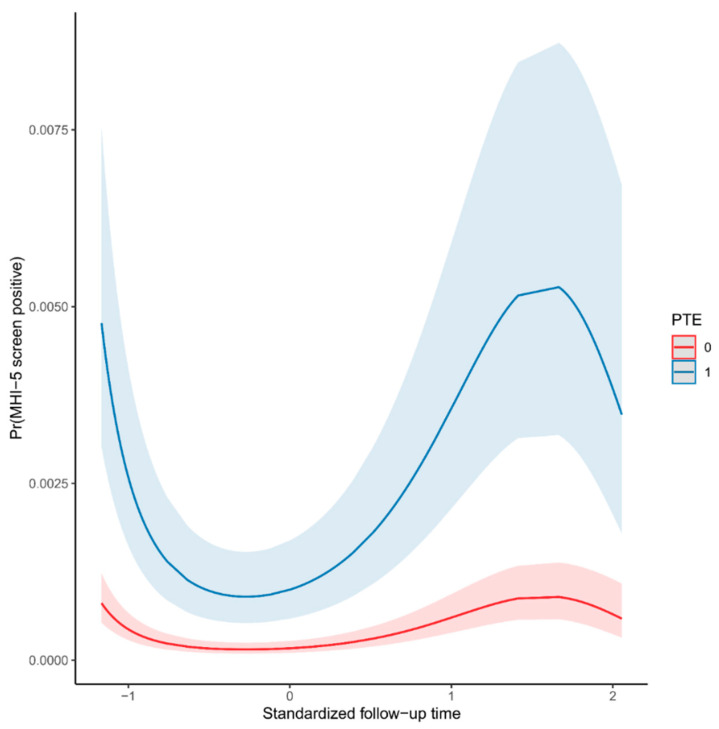
An illustration of Model 2 (Table 2) predictions with zero-random effect. Predicted probabilities (Pr) are for low Mental Health Index 5, i.e., for screen positive MHI-5, when experiencing a potentially traumatic event (PTE = 1) or not (PTE = 0). *X*-axis represents standardized time of answering to questionnaire (mean 0, standard deviation 1), from which the three polynomial terms (“*z*(time)” fixed effects) in Table 2 derived.

**Table 1 ijerph-18-02524-t001:** Average sample characteristics by survey wave.

Wave	0	1	2	3	4	5
Days since 4 June 4 2020	4.0	34.5	68.5	96.0	124.2	159.3
N	4804	2262	2172	1923	1913	1744
At least 1 re-participation ^a^	-	46.8%	63.2%	69.2%	73.1%	75.7%
Woman ^b^	88.6%	89.3%	89.6%	88.8%	89.0%	88.9%
Age	46.7	49.6	48.5	48.2	48.9	48.3
Direct care ^c^	25.7%	23.2%	22.8%	22.8%	23.1%	22.6%
Nurse ^d^	63.0%	62.3%	61.0%	59.9%	59.6%	60.7%
Work changes ^e^	82.4%	39.5%	29.3%	37.1%	40.2%	23.0%
MHI-5 ^f^	72.3	78.3	78.8	76.5	75.0	74.4
MHI-5 ≤ 52	16.7%	9.4%	9.1%	12.0%	14.2%	15.0%
ISI ^g^	7.1	6.3	6.1	6.4	6.6	6.6
ISI ≥ 15	9.5%	5.9%	5.6%	7.5%	7.6%	7.5%
PTE ^h^	27.9%	18.6%	15.9%	13.2%	13.6%	12.8%
PC-PTSD+ if PTE ^h^	23.5%	20.3%	16.2%	17.7%	18.7%	19.8%
PTE1 ^i^	13.0%	8.3%	6.3%	5.2%	4.7%	4.4%
PTE2 ^j^	19.9%	12.0%	9.7%	8.4%	8.8%	7.7%
PTE3 ^k^	2.8%	2.9%	3.4%	2.6%	2.8%	3.0%
PTE4 ^l^	0.8%	1.1%	1.1%	1.0%	1.1%	0.9%

^a^ At least one re-participation to surveys after the zero survey wave. ^b^ Woman or someone not indentifying as man. ^c^ Directly caring for COVID-19 patients at wave 0. Changes by wave attributable to attrition. ^d^ Belonging to nursing staff at wave 0. ^e^ Reporting changes in work due to COVID-19. ^f^ MHI-5 (Mental Health Inderx-5 rating, 0–100 points under 53 refer to psychological distress). ^g^ ISI (Insomnia severity index rating 0–28 points, 15 or over refer to moderate or severe insomnia symptoms). ^h^ Potentially traumatic event (PTE 1–4 combined, questions ^i–l^ below); Primary care Post-traumatic Stress Disorder scale (PC-PTSD-5) three or more yes refer to an elevated risk of PTSD.^i^ Has your work with suspected or confirmed COVID-19 patients included exceptionally disturbing or distressing assignments? ^j^ Have you had strong anxiety due to your own or close one’s risk of contracting serious illness for your work with suspected or confirmed COVID-19 patients? ^k^ Have you or your close one contracted a hospital care requiring severe COVID-19? ^l^ Has a close one to you died of COVID-19?

**Table 2 ijerph-18-02524-t002:** Psychological distress and hospital personnel- multilevel generalized linear multiple regression models.

Fixed Effect	Model 1 (N_obs_ = 13,859, N_per_ = 4457)		Model 2 (N_obs_ = 13,482, N_per_ = 4418)		Model 3 (N_obs_ = 13,449, N_per_ = 4410)		Model 4 (N_obs_ = 13,894, N_per_ = 4465)	
	OR	95% CI	OR	95% CI	OR	95% CI	OR	95% CI
Age ^a^	0.95	0.79–1.15	0.97	0.81–1.17	0.95	0.78–1.18	0.97	0.81–1.16
Woman	1.56	0.83–2.93	1.46	0.78–2.72	1.44	0.78–2.67	1.58	0.84–2.98
Direct care	1.88	1.42–2.49	‒	‒	1.81	1.37–2.39	‒	‒
PTE	‒	‒	5.93	4.52–7.79	6.54	5.00–8.56	‒	‒
log(*r* + 1) *	‒	‒	‒	‒	1.34	1.24–1.44	0.95	0.74–1.23
*z* (time) ^a^	1.8	1.42–2.28	2.08	1.63–2.65	‒	‒	1.97	1.2–3.22
*z* (time) ^2^	3.27	2.74–3.9	3.05	2.54–3.66	‒	‒	3.44	2.52–4.7
*z* (time) ^3^	0.56	0.49–0.65	0.56	0.49–0.66	‒	‒	0.55	0.44–0.67
Random effect:		Variance				Variance		Variance
*σ* _B_	‒	69.6	‒	57.2	‒	52.0	‒	70.9

OR = odds ratio; ci = confidence interval; “‒” = variable not in the model; N_obs_ = number of observations modeled; N_per_ = number of persons modeled. ^a^ Questionnaire answering time-variable standardized (z-score transformed) to mean 0 and variance 1. Squares (^2^) and cubes (^3^) are taken after standardization. * Logarithm of weekly COVID-19 incidence, plus one case to prevent minus infinite log-values. σ_B_ Random-effect variance for between-employee differences in risk of low mental health.

## Data Availability

The data are not publicly available due to privacy restrictions. Please contact the corresponding author.

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
