# Peer review of "COVID-19 Pandemic and Helsinki University Hospital Personnel Psychological Well-Being: Six-Month Follow-Up Results"

_ijerph, 2021, doi:10.3390/ijerph18052524_

Round 1

Reviewer 1 Report

This paper provides an "update" of a continuing survey of psychological symptoms in health professionals managing patients suffering from Covid-19 infections. As such the study aims to provide progress in the collection of data over time and to identify aspects of mental health in these professionals which could provide useful additions to the occupational and personal health care of these individuals. I believe the paper does provide useful information from the data presented and the methodology/data analysis is appropriate for achieving this information. It would be helpful for the authors to outline the limitations of their study, such as the increase in study participation between the initial and follow-up studies, the lack of information regarding personal histories of psychiatric disorders and the efforts made by professionals to seek help with their psychological distress. As "individual variation" was an important factor, further examination of this variable would seem important. 

Author Response

Response: Thank you for the time used to evaluate this ms. We present study participation data in Table 1 and in detail in the supplementary data table A1. We describe HUS Helsinki University Hospital additional psychosocial support services in the third to last chapter of the Introduction.  
Lack of comprehensive psychiatric histories and omnipresence of individual variation are pervasive issues in the field of psychiatric epidemiology, and not specific limitations of this paper. Nevertheless, we agree it is good to point these out for the wider audience. Therefore, we inserted following sentences to the limitations section, with four new references. 
“Liability to psychiatric symptoms strongly reflects individual differences in latent risk factors, including genes and general behavioral traits (e.g., Caspi & Moffit, 2018; Rosenström, et al., 2019), and it is not surprising that the same held in our employee data. A large consortium currently seeks to understand reasons and structure behind individual differences in psychiatric symptoms (Kotov et al., 2017). Future research on mental-health in context of Covid-19 might benefit from a more accurate modeling of psychiatric history and comorbidity. Personality might have a role in the vulnerability, resilience and expressions associated with traumatic events (Jakšić et al. 2012), but we were not able to further study this aspect as no personality questionnaire was used.”

Reviewer 2 Report

I have reviewed the manuscript, titled “COVID-19 Pandemic and Helsinki University Hospital Personnel Psychological Well-being: six-month follow-up results”. It examines psychological distress of the Helsinki University Hospital personnel via an 20 electronic survey monthly since June 2020. The article reports six-month follow-up results of a prospective 21 18 –month cohort study.

The study has many strong points (clear introduction, accurate statistical analysis, constructive discussion).

However, I would like to ask the authors to address some points in order to improve the paper.

Introduction:

1) The Introduction section is very brief; it should contain more information related to psychological well-being (its understanding, underlying mechanisms, theoretical framework).

2) Can you describe more thoroughly the main reasons why frontline but also other HUS personnel were psychologically distressed?

3) What are the relationships between psychological well-being and distress?

4) Please, provide your rationale for examining psychological well-being among hospital personnel (see Krok, D., & Zarzycka, B. (2020). Risk Perception of COVID-19, Meaning-Based Resources and Psychological Well-Being amongst Healthcare Personnel: The Mediating Role of Coping. Journal of clinical medicine, 9(10), 3225)

Method:

4) Can you formulate research questions that would narrow the aim of your study

5) How did you handle missing values in your data? (If any exist)

Discussion:

6) P. 7, line 195-197:  The statement: “In our cohort, individual variation explained much of the psychological distress but  also reported COVID-19 incidence rates in Finland predicted psychological distress and negative changes in sleep”. Can you expand your thoughts? How could the changes in psychological distress affect psychological well-being?

7) Why caring for COVID-19 patients at each survey wave was associated with psychological distress over a six-month time? (p. 7).

8) What mechanisms are responsible for chronic insomnia having adverse effects on mental wellbeing and coping with stress? (page 8).

Author Response

1) The Introduction section is very brief; it should contain more information related to psychological well-being (its understanding, underlying mechanisms, theoretical framework).

Thank you for the time used to evaluate this ms. Information on the Finnish COVID-19 testing and psychiatric research articles has now been added to the introduction from the end of the ms also based on other evaluations. We have now added a sentence on resilience to the end of the ms “Mapping both individual and systemic vulnerability and resiliency factors in this prolonged situation for health care workers should be addressed.”

2) Can you describe more thoroughly the main reasons why frontline but also other HUS personnel were psychologically distressed?

Please see rewritten Introduction. We have now added a sentence of possible factors based on our earlier and other study. Also other data from Finland suggest that the COVID-19 pandemic has caused burden also to health care sectors that have not been directly involved in treating COVID-19 patients. For example, in a survey to the directors of psychiatric services in Finnish health care sectors, personnel well-being at work was reported to be worse than before the pandemic by  46 per cent of directors of adult psychiatric services, 54 per cent of directors of adolescent psychiatric services, and 47 per cent of directors of child psychiatric services, while none of the responders reported any improvement in personnel well-being during the pandemic but we did not add this to our ms because this data is not available at least yet in English https://www.julkari.fi/bitstream/handle/10024/140880/Viikko%206-2021%20-%20Koronaepidemian%20vaikutukset%20hyvinvointiin%20palveluihin%20ja%20talouteen.pdf?sequence=8&isAllowed=y

Study results on anxiety among hospital personnel during the COVID-19 pandemic vary, to clarify that earlier we have now changed the following sentences from Discussion to Introduction “ Among the hospital personnel, exposure, in our cohort also front-line work, is an issue [3]. An Israelian cross-sectional study on hospital worker (N=1570) anxiety during the COVID-19 pandemic revealed that pandemic work in itself (working in a hospital treating COVID-19 patients or in a hospital without COVID-19 patients) did not associate with anxiety. While some individuals are more prone to anxiety symptoms, actually individuals prone to COVID-19 related fear differed from anxiety prone individuals in the German general population.”

3) What are the relationships between psychological well-being and distress?

We agree that lack of distress and psychological symptoms is not well –being in itself. However, the name of the prospective cohort study, which is same in Finnish, was selected for practical purposes in the early days of COVID-19 pandemic, to quickly clarify that this is a prospective cohort study on mental health and ability to function.

4) Please, provide your rationale for examining psychological well-being among hospital personnel (see Krok, D., & Zarzycka, B. (2020). Risk Perception of COVID-19, Meaning-Based Resources and Psychological Well-Being amongst Healthcare Personnel: The Mediating Role of Coping. Journal of clinical medicine, 9(10), 3225)

We have clarified the purpose of the study also based on other evaluations in the Introduction and added the suggested reference.

Method:

4) Can you formulate research questions that would narrow the aim of your study

Response: We have now clarified the end of introduction. We follow psychological distress among HUS personnel during the pandemic as described also in the conclusions, to ensure psychosocial support services needed to support ability to work and function during the pandemic. 

5) How did you handle missing values in your data? (If any exist)

Response: Multilevel models naturally make use of all available data, not just “complete” observations with all assessment time points, which is one of their many benefits. We did not aim to further model the relationship between non-response and key variables in this descriptive study. Missing-data modeling introduces assumptions, which are easier to manage when there is a clear target hypothesis where one strives towards the least biased and most statistically powerful answer. That is, each respondent contributed to our analysis in proportion to data they provided and none who had provided some data was excluded from analysis (as some “complete-case analyses” might do). However, we did not try to model the counterfactual situation of how the results would have looked had we had all the data also from employees that did not answer every time but did some time. We have now clarified this in the end of the Method section. 

Discussion:

6) P. 7, line 195-197:  The statement: “In our cohort, individual variation explained much of the psychological distress but  also reported COVID-19 incidence rates in Finland predicted psychological distress and negative changes in sleep”. Can you expand your thoughts? How could the changes in psychological distress affect psychological well-being? We have now clarified this as follows. Reported increase in COVID-19 incidence rates in Uusimaa district in Finland associated with psychological distress and negative changes in sleep. In an individual level PTEs, other life events, resilience and coping strategies may mediate these changes.

7) Why caring for COVID-19 patients at each survey wave was associated with psychological distress over a six-month time? (p. 7).

In our dataset, potentially traumatic events, PTEs were associated with the screen for low mental health over the entire follow-up. PTEs were statistically significantly more common in front-line personnel in our baseline study. But other data from Finland suggest that the COVID-19 pandemic has caused burden also to health care sectors that have not been directly involved in treating COVID-19 patients. For example, in a survey to the directors of psychiatric services in Finnish health care sectors, personnel well-being at work was reported to be worse than before the pandemic by  46 per cent of directors of adult psychiatric services, 54 per cent of directors of adolescent psychiatric services, and 47 per cent of directors of child psychiatric services, while none of the responders reported any improvement in personnel well-being during the pandemic but we did not add this to our ms because this data is not available at least yet in English https://www.julkari.fi/bitstream/handle/10024/140880/Viikko%206-2021%20-%20Koronaepidemian%20vaikutukset%20hyvinvointiin%20palveluihin%20ja%20talouteen.pdf?sequence=8&isAllowed=y

8) What mechanisms are responsible for chronic insomnia having adverse effects on mental wellbeing and coping with stress? (page 8). We have now added text and references as follows;  Instead, chronic insomnia can have adverse effects on health, mental wellbeing and coping with stress. Long-term cardiovascular, neurological and psychological effects of disrupted sleep are mediated by several mechanisms, for example by low grade inflammation, autonomous hyperarousal, endocrine dysregulation, deficient brain drainage, poor restoration of mitochondrial energy resources, deteriorated memory consolidation and emotion regulation, daytime tiredness, lowered mood, anxiety and maladaptive behavioral responses (Paunio et al 2015, Van Someren et al 2015).

Paunio T, Tuisku K, Korhonen T. Sleep, work and mental health. Psychiatria Fennica 2015;46:54-61.

Van Someren E, Cirelli C, Dijk D,4  Van Cauter E,Schwartz S, Chee M. Disrupted Sleep: From Molecules to Cognition. J Neurosci 2015; 35(41):13889 –13895.

Reviewer 3 Report

Thank you for allowing me to review the paper COVID-19 pandemic and Helsinki university hospital personnel psychological well-being: Six-month follow-up results. I commend the authors for providing insight into front-line worker well-being during the ongoing COVID-19 pandemic. Despite these strengths, I have concerns with the methodological and statistical considerations taken. More specific concerns are outlined here.

  1. I recognize that the purpose of the study is to be descriptive in nature (i.e., reporting important results of a longitudinal data collection effort). However, I believe that study could be improved with a couple of paragraphs in the introduction that reiterate the importance of conducting this study. For example, why do these research questions matter either theoretically or practically? The authors do an adequate job of providing descriptive background for the contextual foundation of the study, but a bit more justification for why this investigation is important would be beneficial.
  2. My reading of the materials section leaves me with a few remaining questions regarding operationalization of focal variables, treatment of missing data, and how data were specified at different levels.
    1. First, for the four questions listed out (e.g., “has your work with suspected or confirmed COVID-19 patients included exceptionally disturbing or distressing assignments?”), are the responses binary Y/N? They’re noted to be “open questions” but it’s unclear if that refers to the typology of the question or the response format. Second, for the well-being measures, it’s my understanding that each scale was transformed to a different scaling and then dichotomized. Is this consistent with how these variables are always treated in research? If there is some inconsistency, it may be worth reconsidering your multi-level models treating these variables as continuous, increasing the degree of variance at the within-person level.
    2. A much more detailed explanation of how missing data was treated is necessary. It appears that pairwise deletion was universally used. While that strategy may be appropriate, it has different implications at different levels of analysis. Was pairwise deletion used for the level two (i.e., between-person level) variables? This lack of clarity makes it difficult to determine the sample size used in the models.
    3. A clearer description of which variables are specified at which level and how those variables were treated is needed. For example, it seems as though MHI-5 and ISI were specified at level 2 (i.e,. between person). How were the between-person component of those scores isolated? Was there a centering procedure? Were raw scores at each wave considered separately? A more detailed explanation is needed. If only the between-person component of these focal variables are considered, comment 2b becomes more important and necessary to address.

Author Response

Thank you for the time used to evaluate this ms. We have now clarified our study question in the end of Introduction and added background information to the Introduction from the end of the ms also based on other evaluations.We follow personnel well-being to ensure ability to work and to assess possible needs in psychosocial support services during the Covid-19 pandemic.

My reading of the materials section leaves me with a few remaining questions regarding operationalization of focal variables, treatment of missing data, and how data were specified at different levels.

The covariates for age and sex varied at “group” level (only across individuals, not within individuals), whereas time and local Covid-19 incidence varied at individual level (depended on exact response time, not just wave). Also, Covid-19 contact and potentially traumatic event varied at the within-employee level because they had been inquired from every respondent at each wave.

 Multilevel models naturally make use of all available data, not just “complete” observations with all assessment time points, which is one of their many benefits. We did not aim to further model the relationship between non-response and key variables in this descriptive study. Missing-data modeling introduces assumptions, which are easier to manage when there is a clear target hypothesis where one strives towards the least biased and most statistically powerful answer. That is, each respondent contributed to our analysis in proportion to data they provided and none who had provided some data was excluded from analysis (as some “complete-case analyses” might do). However, we did not try to model the counterfactual situation of how the results would have looked had we had all the data also from employees that did not answer every time but did some time. We have now clarified this in the end of the Method section. 

First, for the four questions listed out (e.g., “has your work with suspected or confirmed COVID-19 patients included exceptionally disturbing or distressing assignments?”), are the responses binary Y/N? They’re noted to be “open questions” but it’s unclear if that refers to the typology of the question or the response format.  

These four questions were coded yes or no and this is now clarified the Methods, the survey also has open questions on work but symptom rating scales were coded as described in Methods and also in our baseline survey.

Second, for the well-being measures, it’s my understanding that each scale was transformed to a different scaling and then dichotomized. Is this consistent with how these variables are always treated in research? If there is some inconsistency, it may be worth reconsidering your multi-level models treating these variables as continuous, increasing the degree of variance at the within-person level.

A much more detailed explanation of how missing data was treated is necessary. It appears that pairwise deletion was universally used. While that strategy may be appropriate, it has different implications at different levels of analysis. Was pairwise deletion used for the level two (i.e., between-person level) variables? This lack of clarity makes it difficult to determine the sample size used in the models.

Response: Please, see also updated methods. Nothing was “deleted”. Some employees merely had less within-individual data points than other employees and we did not try to ‘guess’ (model) from covariate data what those who withheld data would have responded if somehow forced to do so.

A clearer description of which variables are specified at which level and how those variables were treated is needed. For example, it seems as though MHI-5 and ISI were specified at level 2 (i.e,. between person). How were the between-person component of those scores isolated? Was there a centering procedure? Were raw scores at each wave considered separately? A more detailed explanation is needed. If only the between-person component of these focal variables are considered, comment 2b becomes more important and necessary to address.

Response: Everyone responded to MHI-5 and ISI questions at each wave. Between-person scores are not isolated but modeled as a random effect (as a variance component). These multilevel models are perhaps best thought of as having two kinds of residuals (after modeling fixed effects), an employee-specific group level residual and observation-specific residual not accounted by the former variation. The framework is a widely used one to which we have now added two excellent references at different technical levels of exposition (Gelman & Hill, 2007; Bates, 2010). 

A following is an example on data analysis to clarify previous sentences  (gmf1_c19care <- glmer(mhi5_pos ~ zage + sex + c19_care + zday + I(zday^2) + I(zday^3) + (1|ID), family = "binomial", data = dd)

Gelman, A., & Hill, J. (2007). Data Analysis Using Regression and Multilevel/Hierarchical Models. Cambridge University Press.

Bates, D. M. (2010). lme4: Mixed-effects modeling with R. http://lme4.0.r-forge.r-project.org/lMMwR/lrgprt.pdf

Reviewer 4 Report

1 The subjects were health care workers. There is no information about the social desirability bias, the tendency for research participants to attempt to act in ways that make them seem desirable to other people.

2 The scale used did not indicate reliability and validity.

3 It is better to use the Big Five Personality Questionnaire to further explore the relationship between different doctors' personalities and mental health during the epidemic, which will make this study more bright and contribution. From Freud's personality structure to Cattell RB's Sixteen Personality Factors, no matter what research orientation is adopted, we are trying to construct a personality model that can describe and explain personality characteristics. However, the number and nature of factors included in these numerous personality models are very different, and the consistency is very small. In the 1990s, some psychologists put forward the model of the Five factors of Personality, known as the "Big Five Personality", which has been widely recognized and accepted as a general framework for Personality research. The big five personality traits include extroversion, agreeableness, conscientious, neuroticism and openness to experience.

4 The introduction lacks literature review on some of the key theories and researches on Psychological Well-being and mental health. What's the difference between them? The Symptom Check List 90 is The better way to measure mental health. The Symptom Check List 90 (SCL-90) was used to measure 10 psychological symptom factors, including somatization, obsessive symptoms, interpersonal sensitivity, depression, anxiety, hostility, fear, paranoia, psychosis, and additional factors. Factors were used to reflect the presence of various psychological symptoms and their severity. After each project, the subjects selected 5 grades from 1 to 5 according to the grades of "none, very light, medium, heavy and severe". The subjects selected the appropriate score for each project according to their recent situation and experience. The overall mean, the level of each factor, and the prominent factor were analyzed to understand the scope, presentation, and severity of the patient's problem. SCL-90 can be used for traceability testing.

Author Response

1 The subjects were health care workers. There is no information about the social desirability bias, the tendency for research participants to attempt to act in ways that make them seem desirable to other people.

Thank you for the time used to evaluate this ms. We have now added a sentence to the limitations “We consider social desirability bias unlikely since participation is voluntary.”

2 The scale used did not indicate reliability and validity?

Response: We have now added following sentences with new references and Crohnbach`s alphas to the Method section” In a Finnish general population study, MHI-5 showed good internal consistency and independent factor structure and item characteristics.” (Elovainio M, Hakulinen C, Pulkki-Råback L, Aalto AM, Virtanen M, Partonen T, Suvisaari J. General Health Questionnaire (GHQ-12), Beck Depression Inventory (BDI-6), and Mental Health Index (MHI-5): psychometric and predictive properties in a Finnish population-based sample. Psychiatry Res. 2020 Jul;289:112973.; https://pubmed.ncbi.nlm.nih.gov/32413708/

ISI has been sensitive to change in clinical insomnia studies [8, Järnefelt 2012 a&b]

Järnefelt, H., Lagerstedt, R., Kajaste, S., Sallinen, M., Savolainen, A. & Hublin, C. (2012a). Cognitive behavior therapy for chronic insomnia in occupational health services. Journal of Occupational Rehabilitation, 22, 511−521. Järnefelt, H., Lagerstedt, R., Kajaste, S., Sallinen, M., Savolainen, A. & Hublin, C. (2012b). Cognitive behavioral therapy for shift workers with chronic insomnia. Sleep Medicine, 13, 1238−1246.

Cronbach’s alphas were (in the T0 sample) for the different scales: MHI-5 alpha = 0.895, ISI alpha = 0.895, and PC-PTSD-5 alpha = 0.647. The scales correlate significantly. Pearson’s correlation co-efficient varies from 0.449 to 0.581, suggesting that all the scales measure some aspect of psychological distress, i.e. suggesting concurrent validity.

Referee 4

It is better to use the Big Five Personality Questionnaire to further explore the relationship between different doctors' personalities and mental health during the epidemic, which will make this study more bright and contribution. From Freud's personality structure to Cattell RB's Sixteen Personality Factors, no matter what research orientation is adopted, we are trying to construct a personality model that can describe and explain personality characteristics. However, the number and nature of factors included in these numerous personality models are very different, and the consistency is very small. In the 1990s, some psychologists put forward the model of the Five factors of Personality, known as the "Big Five Personality", which has been widely recognized and accepted as a general framework for Personality research. The big five personality traits include extroversion, agreeableness, conscientious, neuroticism and openness to experience.

Personality might have a role in the vulnerability, resilience and expressions associated with traumatic events (Jakšić et al. 2012), but unfortunately we were not able to further study this aspect as no personality questionnaire was used in this study. We have now clarified this in limitations with a new reference. Unfortunately it is not possible to change the symptom scales during the prospective cohort study (without consulting ethical committee and joint authorities of HUS Helsinki University Hospital who gave permission to conduct this study).

The introduction lacks literature review on some of the key theories and researches on Psychological Well-being and mental health. What's the difference between them? The Symptom Check List 90 is The better way to measure mental health. The Symptom Check List 90 (SCL-90) was used to measure 10 psychological symptom factors, including somatization, obsessive symptoms, interpersonal sensitivity, depression, anxiety, hostility, fear, paranoia, psychosis, and additional factors. Factors were used to reflect the presence of various psychological symptoms and their severity. After each project, the subjects selected 5 grades from 1 to 5 according to the grades of "none, very light, medium, heavy and severe". The subjects selected the appropriate score for each project according to their recent situation and experience. The overall mean, the level of each factor, and the prominent factor were analyzed to understand the scope, presentation, and severity of the patient's problem. SCL-90 can be used for traceability testing.

We have now rewritten Introduction but based on other evaluations we focused on COVID-19 pandemic related issues and studies on mental health. We agree that lack of distress and psychological symptoms is not well –being in itself. However, the name of the prospective cohort study, which is same in Finnish, was selected for practical purposes in the early days of COVID-19 pandemic, to quickly clarify that this is a prospective cohort study on mental health and ability to function.

 We fully agree that SCL-90 would have been a good choice for research purposes, but since it alone has 90 questions we considered that using it repeatedly takes too much time since our questionnaire has also other questions on work.  Unfortunately it is not possible to change the symptom scales during the prospective cohort study (without consulting ethical committee and joint authorities of HUS Helsinki University Hospital who gave permission to conduct this study).

Reviewer 5 Report

Thank you very much for the possibility to review the study titled "COVID-19 Pandemic and Helsinki University Hospital Personnel Psychological Well-being: six-month follow-up results". 

I have read this manuscript with interest and think it adds a lot to the literature, especially with respect to the current pandemic situation. In fact, I think it is important to be able to have more studies on this topic. The authors showed the results that emerged in a six-month follow-up.
Although an excellent job has been done, I feel that the literature on this subject has not been well exposed. In fact, I invite the authors to work more on the introduction highlighting the results of current studies and highlighting more the peculiarities of the Finnish population.

Author Response

Thank you for the time used to evaluate this ms. We have now added information to the Introduction on the Finnish COVID -19 pandemic situation with new references. We have also clarified general information on Finland to the beginning of the Introduction.By February 2021, Finland (population of 5.5 million) has 49,165 confirmed COVID-19 infections and 708 COVID-19 related deaths (median age 84 years).” We have now added studies from Israel, Poland and Germany and a new study from Finland to Introduction, partly from the end of the ms.

Round 2

Reviewer 3 Report

Thank you for the revision. I will try to limit my comments to the points raised in my initial review.

  1. I appreciate that the authors have attempted to clarify their treatment of missing data. I recognize that using pairwise deletion for within-person variables is acceptable in multi-level analyses; however, missingness at level-two (i.e., between-person) poses an issue. This issue could be alleviated if the sample size for the models presented in Table 2 could be provided, both for the number of participants and the number of observations.

Author Response

Thank you again for time used to assess this ms. Sample size for the number of participants and number of observations per model is now provided under the model titles in Table 2 [e.g., as in “Model 1 (Nobs = 13859, Nper = 4457)”]. The revised footnote further explains that “Nobs = number of observations modeled; Nper = number of persons modeled”. We hope this clarified the matter.